


# Aerosol Composition and Extinction of the 2022 Hunga Plume Using CALIOP

Clair Duchamp[1], Bernard Legras[1], Aurélien Podglajen[1], Pasquale Sellitto[2,3], Adam E. Bourassa[4], Alexei Rozanov[5], Ghassan Taha[6,7], and Daniel J. Zawada[4]

[1]Laboratoire de Météorologie Dynamique (LMD-IPSL), Sorbonne Université, CNRS, ENS-PSL, École Polytechnique, Paris, France
[2]Université Paris Est Créteil and Université de Paris-Cité, CNRS, Laboratoire Interuniversitaire des Systèmes Atmosphériques (LISA-IPSL), Créteil, France
[3]Istituto Nazionale di Geofisica e Vulcanologia (INGV), Osservatorio Etneo (OE), Catania, Italy
[4]Institute of Space and Atmospheric Studies, University of Saskatchewan, Saskatoon, SK, Canada
[5]Institute of Environmental Physics, University of Bremen, Bremen, Germany
[6]Morgan State University, Baltimore, MD, USA
[7]NASA Goddard Space Flight Center, Greenbelt, MD, USA

**Correspondence:** Clair Duchamp (clair.duchamp@lmd.ipsl.fr)

**Abstract.** We use the CALIOP (Cloud-Aerosol Lidar with Orthogonal Polarization) instrument to determine the microphysical properties of the stratospheric aerosol plume after the Hunga eruption in 2022, the largest so far after Pinatubo in 1991. In the early stages, low depolarization (<2%) is found everywhere except in patches of high depolarization (up to 35%) detected within the plumes of sulfur compounds up to 3 days after the eruption. As standard CALIOP L2 products are not operational in the case of the Hunga aerosol plume, we implement an iterative method of successive approximations to retrieve extinction profiles, by estimating the aerosol optical depth (AOD) and then the Lidar Ratio (LR). The AOD of the plume at 532 nm is between 0.5 and 1.25 on the first four days, then decreases rapidly and stabilizes at $0.047 \pm 0.011$ for March 2022. LR, initially above 70 sr, is estimated at $48 \pm 6$ sr between late January and late March 2022. Results are compared and validated with the solar occultation instrument SAGE III (Stratospheric Aerosol and Gas Experiment) on board the International Space Station (ISS) and Mie calculations. A comparison with limb-viewing instruments highlights significant quantitative disagreements in extinction and AOD estimates, which we attribute, in part, to the unusual size distribution of the aerosols within the Hunga plume.

## 1 Introduction

After becoming active on December 20, 2021, the Hunga submarine volcano (20.57°S, 175.38°W) produced a spectacular explosive phase, with a Volcanic Explosivity Index (VEI) of ∼6, close to Mount Pinatubo eruption in 1991 (Poli and Shapiro, 2022), on January 15, 2022 by 04:10 UTC (Poli and Shapiro, 2022; Gupta et al., 2022; Yuen et al., 2022). This eruption injected material up to an altitude of ∼58 km by 4:30 UTC, with most of the plume settled between 26 and 34 km altitude afterwards (Carr et al., 2022; Khaykin et al., 2022; Podglajen et al., 2022). In the first hours, volcanic ash, ice and sulfur dioxide were





seen in the stratosphere by geostationary satellite instruments (Himawari-8, GOES-17) before giving way to a plume where

sulfate aerosols (SA) and water vapor coexisted for a few weeks (Sellitto et al., 2022; Carn et al., 2022; Legras et al., 2022). This was due to the rapid conversion of sulfur dioxide, a process enhanced by the abundant presence of water vapor (Millán et al., 2022; Zhu et al., 2022; Sellitto et al., 2024). The injected water vapor was estimated at $146 \pm 5$ Tg, which means an instantaneous 10 % increase of the global stratospheric content (Millán et al., 2022; Khaykin et al., 2022). As a result, the small, spherical, non-depolarizing liquid particles detected by spaceborne and in-situ observations (Legras et al., 2022; Kloss

et al., 2022; Baron et al., 2023) in the days following the eruption formed a long-lived SA plume that circled the globe multiple times, spreading slowly in latitudes between the southern polar region and 20°N (Taha et al., 2022; Legras et al., 2022; Khaykin et al., 2022; Schoeberl et al., 2023b; Sellitto et al., 2024) and still detected at the end of 2023 (Knepp et al., 2024). The particle size distribution (PSD) of the Hunga plume exhibited an effective radius close to $0.4\,\mu\text{m}$ (Khaykin et al., 2022; Baron et al., 2023; Boichu et al., 2023; Duchamp et al., 2023; Asher et al., 2023; Knepp et al., 2024), a mode width $\sigma \sim 1.25$ (Duchamp

et al., 2023; Knepp et al., 2024) and a total density number of 3-5 cm$^{-3}$ (Duchamp et al., 2023). This mode established quickly, within two weeks, and persisted many months. It differs from the background but also from other recent stratospheric eruptions of medium intensity (Wrana et al., 2023) and established much faster than after the Pinatubo eruption (Boichu et al., 2023). Considering these PSD parameters and a $H_2SO_4$ weight proportion of 70%, Duchamp et al. (2023) estimated the total mass of $H_2SO_4$ in SA at $0.66 \pm 0.1$ Tg for the Hunga. This corresponds to an initial $SO_2$ emission of 0.44 Tg in the stratosphere, in

line with modest early 0.4-0.5 Tg estimates (Millán et al., 2022; Carn et al., 2022; Asher et al., 2023) compared to the $18 \pm 4$ Tg of Pinatubo (Guo et al., 2004) and even to the 1.5 Tg of Raikoke (de Leeuw et al., 2021), with a VEI of 4 (Firstov et al., 2020). However, recent studies suggested that the injected $SO_2$ mass burden of the Hunga eruption could have been larger, with 1.0-1.3 Tg $SO_2$ column estimates using satellite data (Sellitto et al., 2024), geological processes (Wu et al., 2025) and modeling (Bruckert et al., 2025). The monomodal distribution assumption used for the retrieval of aerosol PSD with satellite

data could lead, in the event of a second peak in the actual PSD, to a significant underestimation of the total number density and therefore of the total mass (Wrana et al., 2025). Sellitto et al. (2022) showed that water vapor warming impact on the top of atmosphere radiative forcing dominated over aerosol cooling effect, leading to a - very unusual for a volcanic eruption - net warming of the climate system two weeks after the eruption. In the longer term, other studies estimated a dominant aerosol effect slightly decreasing surface temperatures in the Southern Hemisphere in 2022 (Schoeberl et al., 2023a; Gupta et al.,

2025). The Hunga aerosol layer is likely 1.5-10 times more effective at producing a net radiative cooling than those resulting from the El Chichón and Pinatubo eruptions (Sellitto et al., 2025). The aim of this study is to take advantage of the properties of the CALIOP lidar to better characterize the composition of the young plume, and to implement a method for reconstructing the AOD, LR and extinction profiles over the first three months following the eruption, before comparing and validating them with others products.



## 2 Data and Methods

### 2.1 CALIOP & Optical Properties Retrieval

The Cloud-Aerosol Lidar with Orthogonal Polarization (CALIOP) is a two-wavelength lidar on board the CALIPSO satellite (Vaughan et al., 2004; Winker et al., 2010). In this work, we use the L1 v4.51 532 nm total attenuated backscatter averaged in 40 km bins in the horizontal, filtering all profiles which have been contaminated by a low-energy shot. The native vertical resolution is preserved. The molecular backscatter - used to compute the 532 nm attenuated scattering ratio (ASR) - is calculated following Hostetler et al. (2006). The exceptional case of Hunga, due to the height and intensity of its plume (well separated from the tropopause and with good signal-to-noise ratio for night orbits), allows us to retrieve the optical properties of the plume over the first few weeks after the eruption. First, the AOD is estimated from the ASR at aerosol-free altitudes above and below the plume. Then, we calculate the corresponding first guess LR by dividing the AOD by the vertically integrated attenuated backscatter after subtracting molecular backscatter. We use an iterative method of successive approximations to remove the aerosol optical attenuation from above the aerosol layer and retrieve the aerosol extinction profiles with the final LR. The details of the algorithm are given in the next section (2.2). LR depends on some microphysical properties of the observed aerosols such as size, shape, and chemical composition (Fernald et al., 1972). It usually varies from about 5 to 100 sr, with small values often corresponding to large, highly reflective particles and large values suggesting the presence of small absorbing or scattering particles, making it useful for aerosol type classification (Müller et al., 2007; Kim et al., 2018). We present daily results based both on individual orbits (Figure 4) and on daily averages (Figure 5, panel c). For each day, the chosen individual orbit, termed the "most significant orbit of the day", is the one containing the plume with the highest mean ASR above 20 km and over a threshold value of 1.8. Then an average over a selected latitude range is performed inside the plume. On the other side, daily zonal averages are calculated by discarding orbits where a plume is detected but processing is not possible due to a lack of exploitable signal below the plume, that is a range free of aerosols (this case appears progressively from the end of February). Plume free orbits, frequent in the early stage, are counted as zero in the average of the AOD. We limit ourselves to night-time data, as daytime data are more noisy. Measurements in the South Atlantic Anomaly (SAA) are excluded throughout the study, between 90°W and 60°E in the Southern Hemisphere, where the Earth's magnetic field disturbs CALIOP measurements (Noel et al., 2014). Due to solar activity, CALIOP was not operating on 18 January and between 20 and 26 January. This procedure palliates standard CALIOP aerosol L2 products which unfortunately are not suitable for the type of event considered in this study (see Appendix A).

### 2.2 Detailed Optical Properties Retrieval

The lidar equation, after correction of the range effect, is:

$$\beta'(\lambda, z) = \beta(\lambda, z) \exp \left( -2 \int\limits_{z}^{+\infty} \sigma(\lambda, z') \mathrm{d}z' \right) \tag{1}$$



where $\beta$ is the backscatter coefficient, $\sigma$ is the extinction coefficient and $\beta'$ is the retrieved attenuated backscatter stored in the standard CALIOP L1B data product.

The two coefficients are separated into a molecular, a particle and an ozone component: $\beta = \beta_m + \beta_p$ and $\sigma = \sigma_m + \eta\sigma_p + \sigma_{O_3}$ where $\eta$ is the multi-scatter parameter (the $O_3$ backscatter is neglected but not the extinction due to absorption). Molecular and particle LR are respectively $S_m = \sigma_m/\beta_m$ et $S_p = \sigma_p/\beta_p$. According to Hostetler et al. (2006) (p.28-29), we can solve the molecular part using $\beta_m(\lambda, z) = \frac{3}{8\pi k_{bw}(\lambda)}\sigma_m(\lambda, z)$ and $\sigma_m(\lambda, z) = \mathcal{N}_m(z)Q_s(\lambda)$ with $k_{bw}(532 \text{ nm}) = 1.0313$ and $Q_s(532 \text{ nm}) = 5.167 \times 10^{-31} \text{ m}^2.\text{mol}^{-1}$, where $\mathcal{N}_m$ is the molecular number density. $\sigma_{O_3}(\lambda, z) = \mathcal{N}_{O_3}(z)Q_{O_3}(\lambda, T)$, where $\mathcal{N}_{O_3}$ is the ozone number density and $Q_{O_3}$ is the ozone cross-section, dependent on temperature $T$, measured by Serdyuchenko et al. (2014). The attenuated scattering ratio is defined as $\text{SR} = \beta'/\beta_m$ at 532 nm. For a pure molecular profile, it is therefore:

$$\text{SR}_m(z) = \exp\left(-2\int_z^{+\infty}(\sigma_m(\lambda, z') + \sigma_{O_3}(\lambda, z'))\text{d}z'\right)$$

which is always lower than 1. We now define $\Gamma$ as the ratio of SR to $\text{SR}_m$:

$$\Gamma(z) = \frac{\text{SR}(z)}{\text{SR}_m(z)} = \left(1 + \frac{\beta_p(z)}{\beta_m(z)}\right)\exp\left(-2\int_z^{+\infty}\eta\sigma_p(z')\text{d}z'\right) \tag{2}$$

In practice, $+\infty$ is the top of the vertical domain of CALIOP detection, that is 40 km, above which attenuation is assumed to be negligible.

We consider now an aerosol layer which is isolated between a bottom altitude $z_B$ and a top altitude $z_T$, that is there is no aerosol above $z_T$ and there is a layer without aerosols below $z_T$. The AOD of this layer is $\int_{z_B}^{z_T}\eta\sigma_p(z')\text{d}z'$. As $\beta_p(z) = 0$ for $z \leq z_B$ and $z \geq z_T$, the AOD can be obtained as:

$$\text{AOD} = -\frac{1}{2}\ln\left(\frac{\Gamma(z_B^-)}{\Gamma(z_T^+)}\right) \tag{3}$$

where $\Gamma(z_B^-)$ and $\Gamma(z_T^+)$ stand, respectively, for a layer free of aerosol below and above the aerosol layer. In practice, for an aerosol layer with no other aerosols above, $\Gamma(z_T^+)$ can be set to 1. For $z \leq z_B$, we require a layer of $\Gamma$ fluctuating around a minimum which is thicker than 1 km to decide that is can be considered as free of aerosols and usable to measure the AOD. Of course, this is liable to artifacts especially for small AOD and therefore our estimate in such cases must be taken as a lower bound.

It is also necessary to correct $\Gamma$ from the fact that the background is not totally free of aerosols and for calibration issues. Unlike Martinsson et al. (2022) who uses a constant correction factor but for lower altitudes, we estimate it from the attenuated scattering ratio observed in 2021 in a state of CALIOP laser similar to 2022 and for a weakly perturbed stratosphere (besides the remaining aerosols of the 2019-2020 Australian wildfires). For each Hunga case, the 2021 attenuated scattering ratio is averaged at the same latitude and time of the year between 17 km and the altitude of the bottom of the plume. This average is used as a reference for the AOD calculation from the same average made in 2022 under the plume introducing a corrective factor. This factor usually larger than one divides $\Gamma(z_B^-)$. As the background aerosols are possibly replaced and then missing





within the Hunga plume, their effect might not need to be compared and therefore we show results obtained with and without the correction (Figure 5).

Once the AOD is known, and assuming that the particle lidar ratio $S_p$ is constant over the aerosol layer (which means

uniform microphysical properties), we have:

$$C(z) = \frac{\Gamma(z)}{\Gamma(z_T)} = \left(1 + \frac{\sigma_p(z)}{S_p\beta_m(z)}\right)\exp\left(-2\int_z^{z_T}\eta\sigma_p(z')\mathrm{d}z'\right) \tag{4}$$

where $C(z)$ is known from the observations and the molecular properties. For a given $S_p$, this equation can be solved from the top. Since $S_p$ is not known, a determination of $S_p$ and $\sigma_p$ can be done iteratively using:

$$S_p\beta_m(z)(C(z)\exp\left(2\int_z^{z_T}\eta\sigma_p(z')\mathrm{d}z'\right) - 1) = \sigma_p(z) \tag{5}$$

The first iteration neglects aerosol attenuation for a first guess of $S_p$:

$$\eta S_p\int_{z_B}^{z_T}\beta_m(z)(C(z) - 1)\mathrm{d}z = -\mathrm{AOD} \tag{6}$$

This integral can be calculated using the trapezoidal rule over a set of discretized levels. The following iterations calculate

$\sigma_p(z)$ downward from $z_T$ using (5). If the vertical discretization is defined, counting from the top, with a step $\Delta z$ and if $z_T$ and $z_B$ are, respectively, the levels $i$ and $j$ with $j > i$, the first part of the iteration calculate the extinction and the attenuation factor $d$ by initializing them as:

$$
\begin{aligned}
d(i) &= 1 \\
\sigma_p(i) &= S_p\beta_m(i)(C(i) - 1)
\end{aligned}
$$

and then for $k$ in $[i+1, j]$:

$$
\begin{aligned}
d(k) &= d(k-1)\exp\left(2\eta\sigma_p(k-1)\Delta z\right), \\
\sigma_p(k) &= S_p\beta_m(k)(C(k)d(k) - 1)
\end{aligned}
\tag{7}
$$

The second part of the iteration recalculates $S_p$ as:

$$S_p^* = \frac{\mathrm{AOD}}{\eta}\left[\sum_{k=i}^{j}\beta_m(k)(1 - C(k)d(k))\Delta z\right]^{-1}. \tag{8}$$

where the sum is performed using the trapezoidal rule. The new estimate of $S_p$ is then obtained by combining the previous estimate with (8) as:

$$S_p \leftarrow \frac{1}{2}(S_p + S_p^*) \tag{9}$$





The iteration is made until $S_p$ converges. The convergence is greatly accelerated by (9) and is in practice obtained in less than 8 steps for all investigated cases.

In this study, we neglect multiple scattering by setting $\eta = 1$.

Figure 1 shows an example (Cloud C1 from January 17) of the procedure described above from the attenuated scattering ratio (1a, 1b, 1c; computed with the attenuated backscatter stored in the L1B files) to the aerosol extinction $\sigma_p$ (1g). The standard deviation from the mean (1j, 1j, 1l) is chosen to calculate the error bars for the individual cases. For this specific case, we calculate an AOD of $1.24 \pm 0.13$ and then a LR of $70.9 \pm 2.5$ sr.

## 2.3    SAGE III/ISS

The Stratospheric Aerosol and Gas Experiment III (SAGE III) instrument, on board the International Space Station (ISS), has been providing measurements of solar occultations since June 2017 (Cisewski et al., 2014). The instrument provides aerosol extinctions at nine wavelengths from 384 to 1543 nm in 0.5 km vertical steps between 0 and 45 km altitude. The instrument observes about 15 sunrises and 15 sunsets per day with a latitudinal range which varies depending on the period of year. The

aerosol extinctions are retrieved as residuals of a spectral multilinear fit for $O_3$ and $NO_2$ but do not require any size distribution assumptions unlike instruments with limb-scatter geometry like OSIRIS (Bourassa et al., 2007) and OMPS-LP (Loughman et al., 2018). Due to its low geographical sampling, SAGE III only began to observe the Hunga SA plume from February 2022, with several profiles on only 3 days (February 7, 15 and 16), whereas the plume was detected for most of the days in March 2022. For this study, we use the version 5.3 of the SAGE III/ISS level 2 solar aerosol product.

## 2.4    OMPS-LP

The Ozone Mapping and Profiler Suite Limb Profiler (OMPS-LP) instrument, on board the Suomi-NPP satellite, provides aerosol extinction profiles from measurements of the scattered solar radiation in 1 km vertical steps. In this study, we use the 745 nm band of the v2.1 aerosol extinction retrieval algorithm developed by NASA (Taha et al., 2021) and two other OMPS-LP products : the 745 nm band of the v1.3 aerosol extinction product developed by the University of Saskatchewan

(USASK) (Bourassa et al., 2023) and the 869 nm band of the v2.1 aerosol extinction coefficient product developed by the Institute of Environmental Physics (IUP) after a conversion to 745 nm (Rozanov et al., 2024). Of the three slits separated by 250 km at the tangent point designed to observe the atmosphere (Jaross et al., 2014), we use the middle one as it has better straylight performance and pointing knowledge. The extinction is averaged daily over all orbits of that day outside the SAA zone and after a horizontal interpolation to a standard latitude grid of $1.1°$ resolution that corresponds to the mean resolution

of OMPS-LP in the considered range of latitudes.

## 2.5    OSIRIS

The Optical Spectrograph and InfraRed Imaging System (OSIRIS), on board the Odin satellite, measures vertical profiles of spectrally dispersed, limb scattered sunlight from the upper troposphere to the lower mesosphere (Llewellyn et al., 2004). In




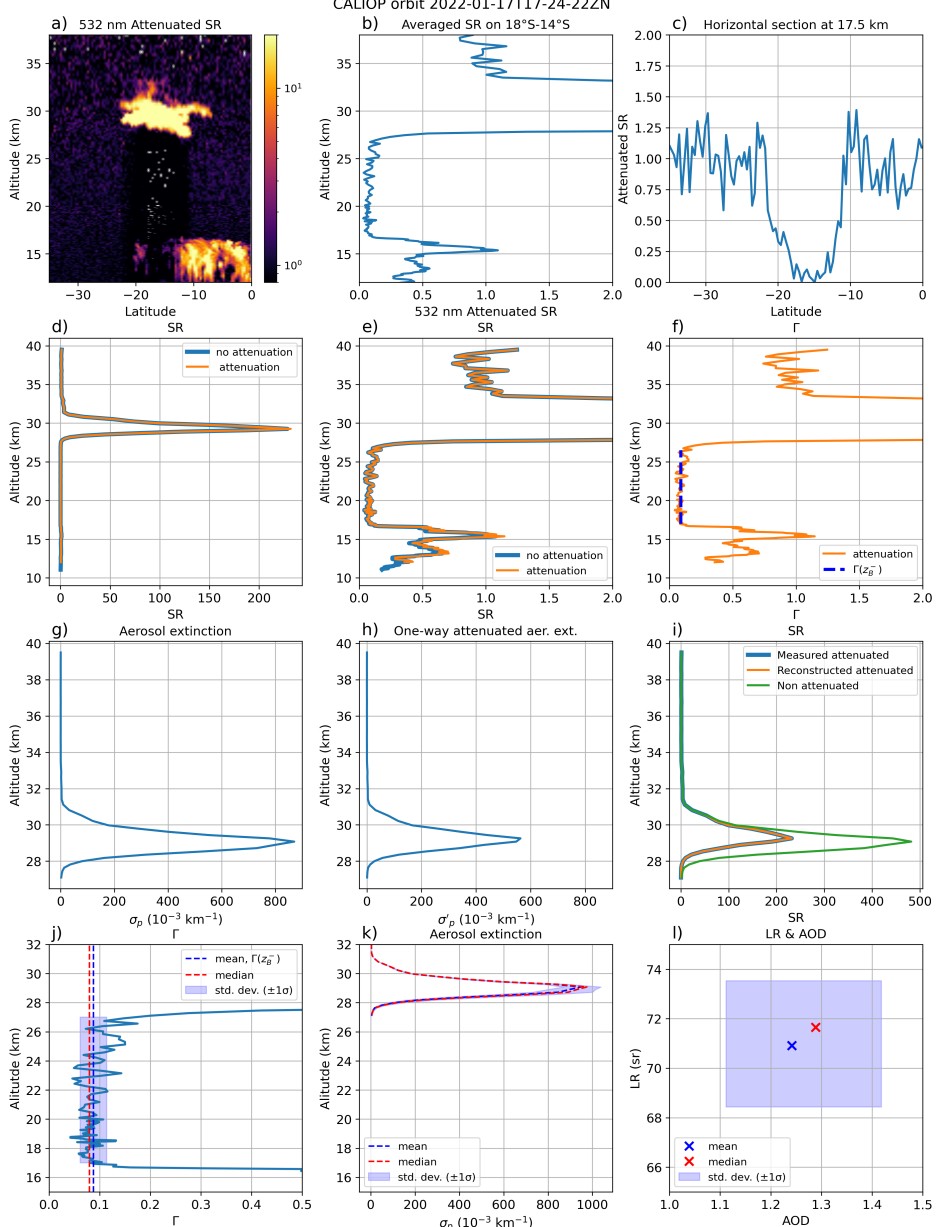

**Figure 1.** (a) CALIOP 532 nm attenuated scattering ratio SR from orbit 2022-01-17T17-24-22-ZN on January 17, 2022. (b) Same as (a) but averaged between 18°S and 14°S. (c) Horizontal section of (a) at 17.5 km altitude. (d) Blue curve: same as (b) (SR, no molecular attenuation). Orange curve: (b) taking into account molecular attenuation including $O_3$ transmission ($\Gamma$, eq. 2). (e) Same as (d) with a reduced scale to better see variations in the vicinity of 1. (f) Orange curve: Same as (e). Blue dotted line: corresponds to $\Gamma(z_B^-)$. (g) Aerosol extinction $\sigma_p$ retrieved together with $S_p$. (h) One-way attenuated aerosol extinction $\sigma_p'$ which is $\sigma_p/\sqrt{d}$. (i) Blue curve: measured CALIOP attenuated scattering ratio. Orange curve: reconstructed attenuated scattering ratio from retrieved $\sigma_p$, $S_p$ and $d$. Green curve: reconstructed non attenuated scattering ratio from retrieved $\sigma_p$ and $S_p$. (j) Mean ($\Gamma(z_B^-)$, blue dotted line), median (red dotted line) and standard deviation (blue area) of $\Gamma$ calculated below the aerosol layer. (k) and (l) are respectively the corresponding aerosol extinction $\sigma_p$ and LR & AOD from $\Gamma$ values (j).



this work, we use v7.3 of the level 2 Odin/OSIRIS stratospheric aerosol extinction coefficient profile at 750 nm (Rieger et al.,
2019) using a multiwavelength retrieval that improves the accuracy of the extinction product by reducing sensitivity to the
unknown PSD in the inversion. This product provides aerosol extinction coefficient in 1 km vertical steps between 0.5 and 44.5
km altitude.

## 2.6 Mie Calculations

We convert CALIOP AOD results from 532 nm to 756 nm and obtain the theoretical LR as a function of the PSD (see
Section 3.2) by calculating extinction and backscatter coefficient using miepython (see *Data availability*), a python code im-
plementing the Mie theory according to Wiscombe (1979). As input, we use fixed real part refractive indices from the GEISA
database (Armante et al., 2016), for a temperature of 215 K and a $H_2SO_4$ weight proportion $w_s$ of 70% (Tabazadeh et al.,
1997) considering that the rest of the liquid droplet is water (Biermann et al., 2000), that is $\{n_{532}, n_{756}\} = \{1.439, 1.438\}$. As
the SA have a very low absorption in the shortwave spectral range (Palmer and Williams, 1975), we fix the imaginary part of
the refractive index to $10^{-6}$. Unimodal lognormal size distribution is assumed for the Hunga SA plume following the results of
Duchamp et al. (2023). From this article (Fig. 2), we derive median radius $r_m = 0.35\,\mu m$ and mode width $\sigma = 1.25$, producing
a conversion factor from 532 to 756 nm of 0.815.

## 3 Results

### 3.1 Plume Composition

The Hunga eruption injected volcanic ash, $SO_2$ and water vapor into the stratosphere (Millán et al., 2022; Legras et al., 2022).
The ash/ice component rapidly dissipated, likely due to sedimentation (Sellitto et al., 2022; Khaykin et al., 2022). A remaining
ash or ice plume was observed by CALIOP on January 15 between 33 and 39 km altitude with a high depolarization ratio (over
30%, see Figure 2a), then it was observed again on January 16 by CALIOP and by a ground-based lidar over Reunion island on
January 20 (Baron et al., 2023). After that date, it could be followed only by OMPS-LP (Taha et al., 2022). The main plume of
sulfur compounds was revealed by Himawari 8 and IASI observations within a day of the main eruption suggesting two separate
aerosol clouds named C1 and C2 (Sellitto et al., 2022; Legras et al., 2022), moving westward, which were also observed by
CALIOP (Figure 2b) and by UV satellite instruments (Carn et al., 2022). These observations suggested that the conversion of
$SO_2$ into sulfates was very fast for the two clouds, especially for the moistier C1 (Sellitto et al., 2022). Both clouds exhibited
a high scattering ratio with very low depolarization in their bulk indicating the predominance of small spherical particles.
Nevertheless, two small patches were observed, in the lower part of C2, of higher depolarization close to 35% with a large
color ratio (CR), which suggested remaining ice crystals or ash.

Two successive orbits from January 17 (Figure 2c, 2d) revealed, in turn, the two clouds with spots of maximum depolar-
ization of 4-5% which could have corresponded to dissipating ice or falling ash. Some other pockets of depolarization were
observed in other aerosol clouds below 18 km altitude (not shown). The higher depolarization spots also exhibited a larger CR







**Figure 2.** 6 CALIOP night orbits on January 15 (a), 16 (b), 17 (c, d) and 19 (e, f). For each panel, from left to right, are represented the 532 nm ASR, the 532 nm depolarization ratio (DR in %, orthogonal channel/total) and the 1064 nm / 532 nm color ratio (CR, 1064 nm backscatter coefficient is not available above 30 km).



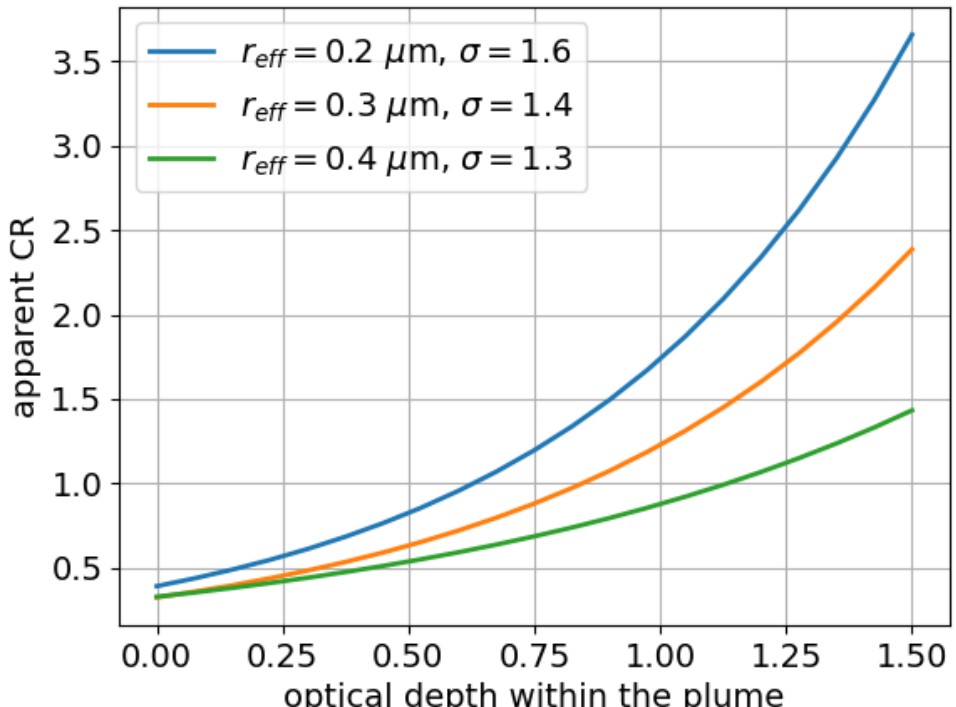

**Figure 3.** Theoretical CR (1064/532 nm) calculated with a Mie code (See Section 2.6) as a fonction of the partial optical depth from the top of the plume for possible PSD assumptions in the early plume ($r_{\mathrm{eff}} = 0.2\,\mu\mathrm{m}$ with $\sigma = 1.6$ in blue, $r_{\mathrm{eff}} = 0.3\,\mu\mathrm{m}$ with $\sigma = 1.4$ in orange and $r_{\mathrm{eff}} = 0.4\,\mu\mathrm{m}$ with $\sigma = 1.3$ in green).

compared with other plume sections, consistent with the dissipation of solid particles (Prata et al., 2017). At this stage, the aerosol index from UV satellite instruments did not reveal any ash in the two clouds (Taha et al., 2022).

The two clouds were seen again on January 19 (Figure 2e, 2f) with depolarization everywhere below 2%, before the interruption of CALIOP operations. During this interruption, the two clouds elongated under the zonal shear (Legras et al., 2022) and were no longer easily distinguishable after January 27 when CALIOP resumed operation. All the subsequent orbits showed

very low depolarization and CR, consistent with a plume containing only small spherical particles.

We note an increase in the CR towards the bottom of plume, by a factor 2 to 4 (Figure 2, panels c, d). Although this feature could reflect an actual change in the particle size, it is quantitatively consistent with differential attenuation between the two wavelengths keeping a constant size distribution and we interpret it as such.

Figure 3 shows that the apparent CR calculated from attenuated backscatter increases with the partial optical depth from the

top of the plume and so may vary significantly across the plume when the AOD (that is the total optical depth) is large, even with uniform optical properties. For instance, this can be seen for cloud C1 on January 17, with theoretical CR values ranging





from 0.4 to 1.2 and 1.75 (green and orange PSDs respectively) from top to bottom of the plume (CALIOP AOD value of 1.24) in agreement with the observed CR values provided by CALIOP (Figure 2, panel d).

## 3.2 Optical Properties

Figure 4 shows the temporal evolution of the 532 nm AOD (a) and LR (b) along selected orbits over the first few weeks of plume dispersion in the stratosphere. Early AOD values were :

1. January, 16 : $AOD_{C1}$ = 0.51 ± 0.07 ; $AOD_{C2}$ = 0.59 ± 0.08

2. January, 17 : $AOD_{C1}$ = 1.24 ± 0.13 ; $AOD_{C2}$ = 1.01 ± 0.12

3. January, 19 : $AOD_{C1}$ = 0.62 ± 0.08 ; $AOD_{C2}$ = 0.89 ± 0.09

Using a similar method for January 16, Sellitto et al. (2022) obtained values of ≈ 0.55 for C1 and ≈ 0.60 for C2. On January 19, a peak in AOD, reaching ≈ 1 at 500 nm and lasting about 12 hours, was recorded by ground-based photometric observations from the AERONET station of Learmonth, western Australia (Boichu et al., 2023). CALIOP measurements then were interrupted but a 532 nm ground-based lidar located on Reunion Island (21.08°S, 55.38°E) provided Hunga plume AOD maximum values of 0.84 and 0.55 respectively on January 21 and 22 (Baron et al., 2023) also detected in AERONET stations measurements on Reunion Island (Boichu et al., 2023). The values of the ground measurements were consistent with the CALIOP values recorded before (0.89 and 0.62 for January 19) and after the interruption (0.15 and 0.24, respectively, for January 27 and 28). AOD values rapidly decreased during the first few weeks after the eruption. In March 2022, the AOD curve tended to stabilize at a monthly value of 0.047 ± 0.011 (error calculations with independent data, see Appendix B).

The LR calculations fluctuated mainly between 40 and 60 sr between January 27 and March 25, with a slightly downward trend over time, resulting in a mean value of 48 ± 6 sr over this period. LR values, like AOD values, were highest during the four days following the eruption, when the plume was compact and localized, with values between 60 and 80 sr for C1 and C2. The Reunion Island lidar measured early LR within the Hunga peak aerosol plume between 68 and 75 sr at 532 nm between January 21 and 23 (Kloss et al., 2022; Baron et al., 2023).

Some properties of the Hunga plume were comparable to those produced by other volcanic eruptions. The two-month average of the LR was similar to the 48 sr value at 532 nm and 16 km altitude observed for the Nabro SA plume in June 2011 (Sawamura et al., 2012) and to the unconstrained assumed value 50 sr used in CALIOP L2 retrieval. One hypothesis to the high early LR values is that they resulted from the presence of residual ash (see Section 3.1) as ash could tend to increase the LR (Vaughan et al., 2021). Prata et al. (2017) found mean LR of 69 ± 13 sr, at 532 nm using CALIOP at a top mean altitude of 12.5 km for the Puyehue-Cordón Caulle eruption in June 2011. Lopes et al. (2019) retrieved LR values of 76 ± 27 sr between altitudes of 18 and 19.3 km for the Calbuco eruption in April 2015. Both of these two eruptive plumes were characterized by a relatively large amount of ash. Another hypothesis is that the effective radius of the sulfate particles was smaller while maintaining a narrower distribution (Figure 6).

In order to test our method, we compare the results with extinction profile data from passive spaceborne sensors at coinciding measurement points. Figures 5a and 5b show collocated individual profiles of CALIOP, SAGE III and OMPS-LP on February

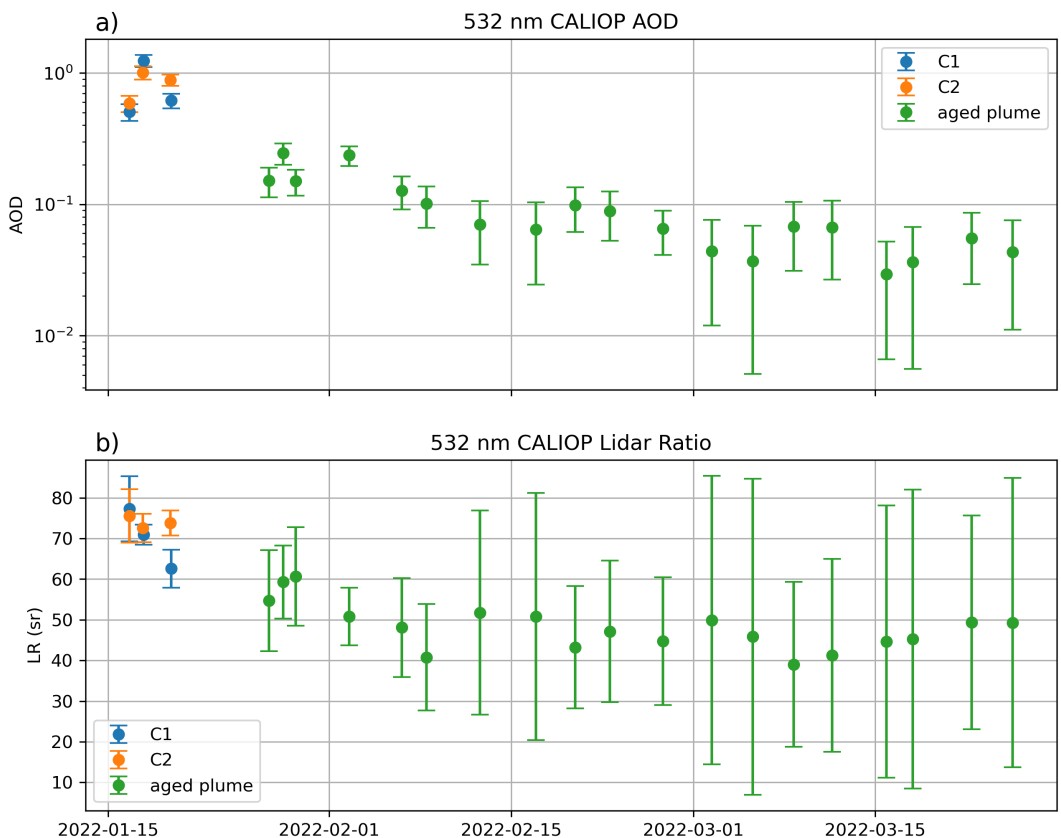

**Figure 4.** (a) Evolution of the AOD retrieved from CALIOP 532 nm attenuated backscatter between January 16 and March 25, 2022. Blue and orange circles correspond respectively to C1 and C2 early clouds. When we can no longer easily distinguish these two plumes, from January 27 onwards, we use the green circles to represent the case of the most significant orbit of the day over all the night orbits of the day outside the SAA zone (see Section 2.1). (b) Same as (a) for LR. The error bars come from standard deviation to the mean calculations (see Figure 1j).



15 near the equator and March 20 near 20°S. Such coincidences are rare events over the period due to the scarce spatiotemporal sampling of SAGE III profiles (see Section 2.3). In both cases, the very good agreement between CALIOP and SAGE III extinction profiles is striking in terms of profile shape and values. Data from the three OMPS-LP products agree regarding the altitude of the extinction peak, but significantly underestimate its magnitude compared to direct extinction measurements from SAGE III, which we take as a reference (see Section 2.3). The discrepancy arises from the limited sensitivity of OMPS-

LP in the early stages, as its long limb path prevents it from seeing into the plume effectively and the retrieval dependency on particle size assumptions (Bourassa et al., 2023). The gap becomes smaller with time as the plume gets diluted, as seen by comparing March 20 with February 15. The IUP product exhibits higher peak values than the other OMPS-LP products, bringing it closer in magnitude to SAGE III and CALIOP. Below the peak, the larger extinction in the NASA product is due to an artifact originating from a convergence problem in the algorithm that will be corrected in the next version.

In Figure 5c, we compare the evolution of AOD daily zonal means with the same products of Figures 5a and 5b, adding the observations from the OSIRIS instrument (see Section 2.5). The figure also shows the SAGE III average over March 2022. The SAA longitudes are excluded for all instruments but SAGE III. Until early March, all curves show the same variations with two wide gaps corresponding to two periods where the bulk of the plume was located in the SAA region. CALIOP values for the January 16 are lower than those for January 17 and 19 due to sampling (also visible in Figure 4). Over January and early

February, the 756 nm CALIOP converted curve is above the others with the OMPS-LP NASA product providing the closest match — likely overestimated due to the artifact below the peak. In the second half of February, the converted CALIOP curve and the OMPS-LP curves agree within the uncertainty which ranges between 15% and 50% for CALIOP (see Figure B1) and, with OMPS-LP NASA, above all other curves. After March 1$^{st}$, when the dispersion removes the SAA dependency, CALIOP AOD falls below the OMPS-LP NASA data which remains at constant level and matches SAGE III. This may be due to the

difficulty to measure AOD on many CALIOP orbits as the plume gets diluted, weakening the average signal, while the higher sensitivity of OMPS-LP still produces reliable results. However, the CALIOP curve falls within the other OMPS-LP products and, when the AOD is large (panel b), individual profiles still match SAGE III. Based on these results, we fit an exponential-decay curve to CALIOP before March and SAGE III data in March, suggesting an overall decay of the AOD with an e-folding time of 19.3 days. The use of correction factor (see Section 2.2) for CALIOP data does not significantly change the results

(orange vs. ochre curves). The OMPS-LP values of AOD, and differences between the three products, are consistent with existing literature (Sellitto et al., 2022; Taha et al., 2022; Khaykin et al., 2022; Bourassa et al., 2023) and their differences with SAGE III are on average smaller than with individual profiles for which the low sensitivity bounds the values. The AOD curve from OSIRIS is far below the others with values ∼0.01. The substantial low bias in OSIRIS aerosol extinction during the Hunga time period has been previously noted by Rozanov et al. (2024). It is believed to be caused by particle size distribution

assumptions in the retrieval that are amplified relative to OMPS-LP by the higher solar zenith angles under which OSIRIS operates. The unusual aerosol size distribution of the Hunga plume leads to the significant quantitative disagreements in AOD estimates from scattered light measurements due to the different size distribution assumptions, as well as the sensitivity of the instrument.



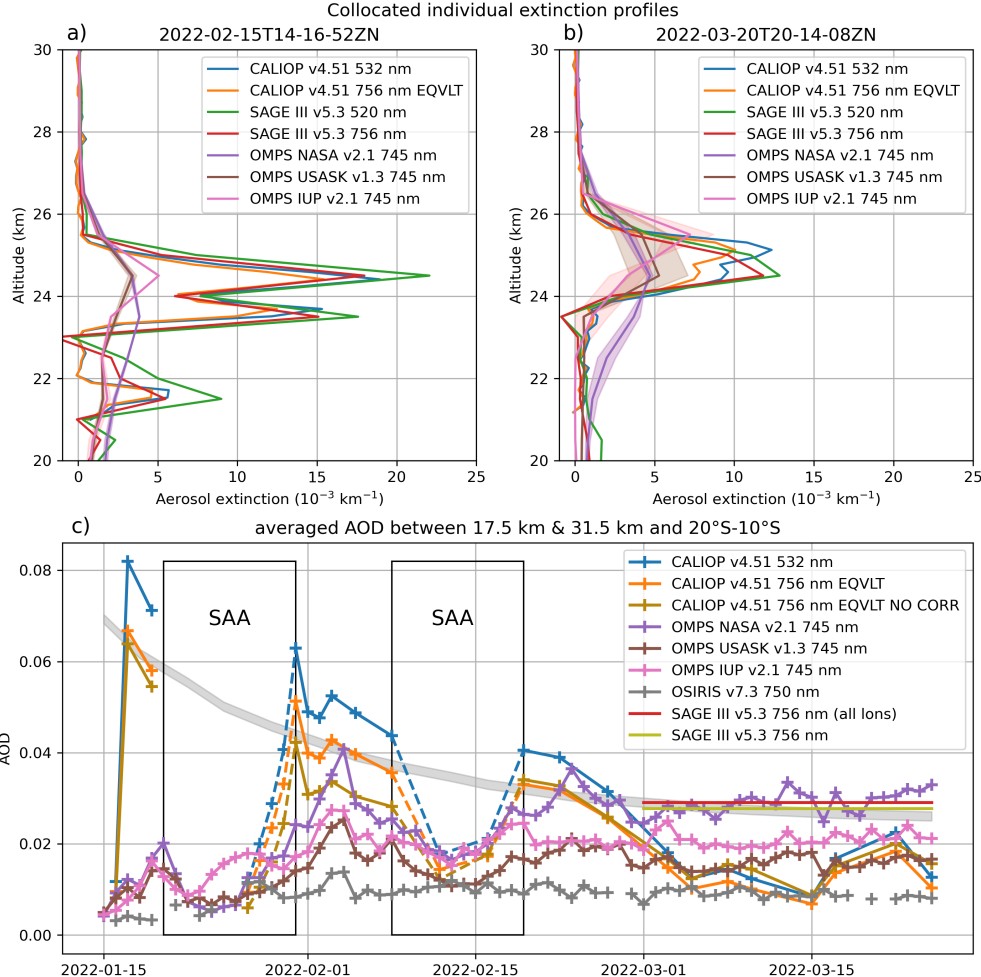

**Figure 5.** (a) Collocated profiles from CALIOP, SAGE III and OMPS-LP on February 15. CALIOP profile is averaged between 4°S and 2°S at 178.5°E and 14:43 UTC. SAGE III profile is measured at 3.2°S, 178.4°E and 18:17 UTC. OMPS-LP profile is averaged between 4°S and 2°S at 171.7°W and 00:54 UTC. (b) Collocated profiles from CALIOP, SAGE III and OMPS-LP on March 20. CALIOP profile is averaged between 21.3°S and 19.3°S at 87.2°E and 20:40 UTC. SAGE III profile is measured at 20.3°S, 88.1°E and 00:15 UTC (March, 21). OMPS-LP profile is averaged between 21.3°S and 19.3°S at 95.5°E and 07:16 UTC. CALIOP night orbits and OMPS-LP day orbits will always have a time lag of about half a day. To compensate for this, we choose a different spatial band between CALIOP and OMPS-LP, taking into account the ERA5 angular velocity using panels (d) and (f) from Fig. 1 in Legras et al. (2022). Shading corresponds to the two OMPS-LP profiles that are averaged over the 2° latitude band. (c) Comparison of daily averaged AOD retrieval excluding profiles in the SAA longitudes from CALIOP (see Section 2.1) with several products from passive sensors measuring extinction (SAGE III, OMPS-LP and OSIRIS). Calculations are made between 17.5 and 31.5 km altitude and averaged between 20°S and 10°S. The ochre curve represents CALIOP AOD without correction (see Section 2.2). The OMPS-LP products contain a large amount of data per day, whereas SAGE III and OSIRIS provide much less data due to their measurement geometry. This is why OSIRIS is missing a few days, and the SAGE III values (red, accounting all longitudes, and green, excluding SAA longitudes) for March are monthly averages. The black boxes represent time periods when the main part of the plume is in the SAA zone. For all the three plots, CALIOP 756 nm is converted from CALIOP 532 nm using Mie calculations (see Section 2.6).

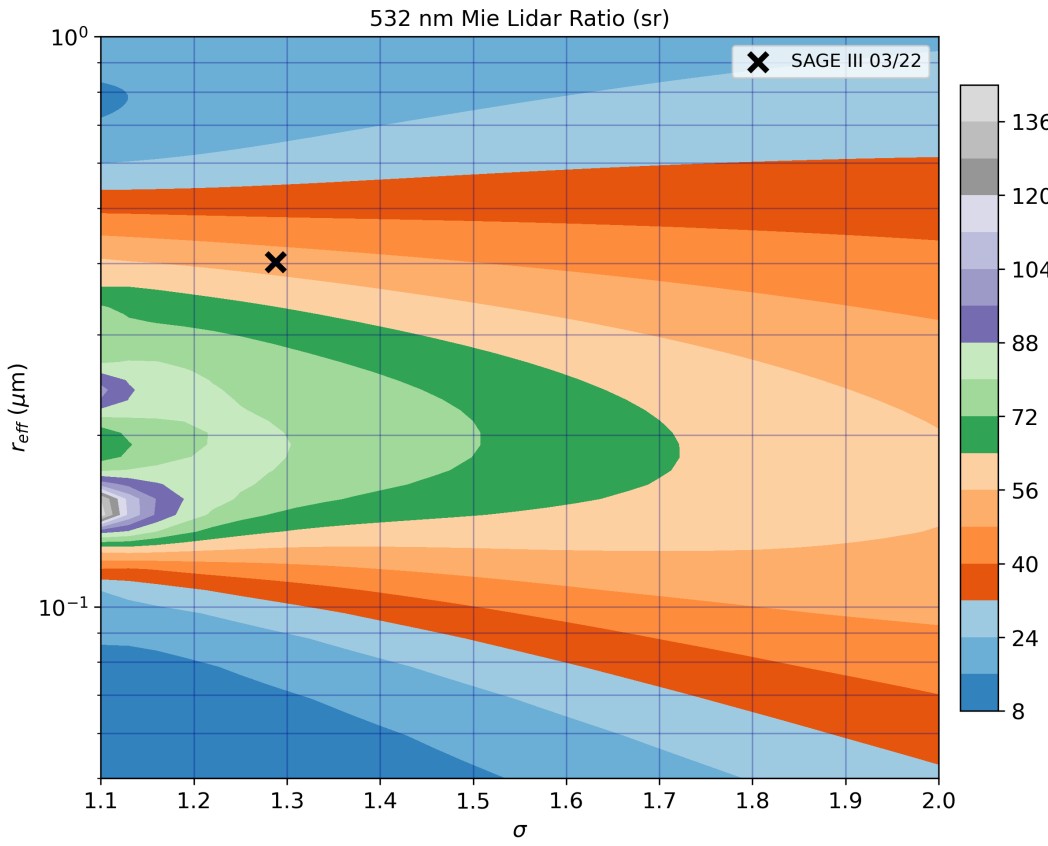

**Figure 6.** Theoretical LR calculated with a Mie code (see Section 2.6) as a function of the effective radius $r_{\mathrm{eff}}$ and mode width $\sigma$ of a lognormal distribution. The black cross represents the mean values from SAGE III of $r_{\mathrm{eff}}$ and $\sigma$ - respectively 0.40 μm and 1.29 - inside the plume in March 2022 between 30°S and 10°S taken from Figure 2 in Duchamp et al. (2023).

Assuming the Hunga plume consists of small spherical particles, the observed LR is liable to be compared with Mie theo-
retical calculations. Figure 6 shows the numerically calculated 532 nm LR as a function of PSD parameters for a lognormal distribution. LR is very sensitive to PSD variations for $r_{\mathrm{eff}}$ ranging from $\approx 0.1$ to 0.5 μm and for low $\sigma$ values. The black cross, representing the average $r_{\mathrm{eff}}$ and $\sigma$ within the plume observed by SAGE III for March 2022 in the tropics, corresponds to a theoretical LR of $\approx 52$ sr, well within the range of values determined by our CALIOP estimate, enhancing the overall confidence in our results.

**4 Conclusions**

The CALIOP lidar monitored the formation and evolution of Hunga aerosol plumes in the stratosphere from the day of the eruption to the end of its mission in mid-2023, fuelling numerous studies (Sellitto et al., 2022; Legras et al., 2022; Khaykin



et al., 2022). The quality and high resolution of its night-time measurements, with a good signal-to-noise ratio, provides several insights into the composition of the plume. First, the detection of highly depolarized pockets in the early plumes suggests the presence of remaining ice crystals or ash. The optical properties of aerosols for well-separated plumes in the first few days after the eruption, as well as for the uniform, dispersed plume in the following weeks, could then be estimated. The AOD of the plume is between 0.5 and 1.25 on the first four days, then, values decrease and stabilize at $0.047 \pm 0.011$ for March 2022. LR values, initially above 70 sr, is estimated at $48 \pm 6$ sr over the January 27 - March 25 period. The data are compared and validated on individual profiles with direct extinction measurements by solar occultations from SAGE III when it detects the plume, and with Mie theoretical calculations. CALIOP values are higher than those of the various OMPS-LP products. In terms of daily averaged AOD, the NASA product shows the closest match during the early, concentrated stage of the plume (until March 2022). For individual extinction profiles, the IUP product provides slightly better agreement than the other OMPS-LP products.

*Data availability.* CALIPSO Lidar Level 1B profile data, V4-51 are available at: https://doi.org/10.5067/CALIOP/CALIPSO/CAL_LID_L1-Standard-V4-51. SAGE III/ISS L2 Solar Event Species Profiles V053 data set is available from https://doi.org/10.5067/ISS/SAGEIII/SOLAR_NetCDF4_L2-V5.3. OMPS-LP V2.1 aerosol extinction coefficient data product from the IUP is available at https://www.iup.uni-bremen.de/DataRequest. OMPS-NPP L2 LP Aerosol Extinction Vertical Profile swath daily 3slit V2 data from NASA are available at https://doi.org/10.5067/CX2B9NW6FI27. OMPS-NPP L2 LP USask Aerosol Extinction Vertical Profile swath daily V1.3 data are available at sftp://odin-osiris.usask.ca. OSIRIS L2 V7.3 data are available at https://arg.usask.ca/docs/osiris_v7/. miepython is a free package available from https://miepython.readthedocs.io/en/latest/ and https://doi.org/10.5281/zenodo.11135148. We used version 2.5.4 in this study. AERIS has provided access to the GEISA database at https://geisa.aeris-data.fr.

## Appendix A: NASA Standard CALIOP L2 Product

NASA has developed standard L2 level products derived from the attenuated backscatter and offers orbital files of type '05kmAPro' (https://doi.org/10.5067/CALIOP/CALIPSO/CAL_LID_L2_05kmAPro-Standard-V4-51) and '05kmALay' (https://doi.org/10.5067/CALIOP/CALIPSO/CAL_LID_L2_05kmALay-Standard-V4-51) containing variables we retrieve as part of our study such as the 532 nm extinction coefficient, the 532 nm stratospheric AOD (SAOD), the initial and final 532 nm LR and associated quality flags, among others (Young et al., 2008).

The Figure A1 presents NASA L2 data for 3 orbits containing the Hunga SA plume on 17 (a, C2), 19 (b, C1) and 27 January (c). Most of the time, the LR is not constrained and its value does not change from that set at the outset, which is 50 sr. When the value is constrained, the algorithm seems to produce abnormally low LR values under 50 sr and sometimes close to 0 sr. In comparison, the values we obtain in the plume are respectively $72.5 \pm 3.5$ sr, $62.6 \pm 4.6$ sr and $54.7 \pm 12.4$ sr for orbits from panel (a), (b) and (c). This can lead to important differences for the retrieval of extinction and SAOD. For instance, mean NASA SAOD is around 0.4 (maximum around 0.7) while we find 1.01 in averaging inside the plume for orbit from panel (a).

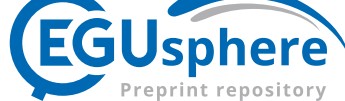

**Figure A1.** 3 CALIOP night orbits on January 17 (a, C2), 19 (b, C1) and 27 (c). For each panel, from left to right, are represented the 532 nm extinction coefficient, the 532 nm LR (extinction/backscatter), whether the LR is constrained (1) or not (0) to retrieve the extinction and the 532 nm stratospheric AOD (SAOD).





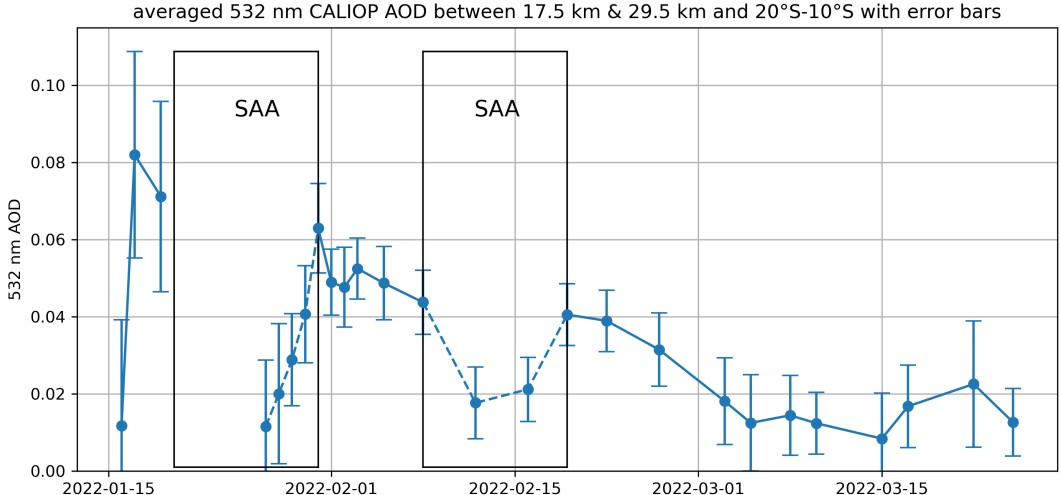

**Figure B1.** Same as Figure 5c with error bars for CALIOP 532 nm daily AOD.

## Appendix B: Error Calculation of an Averaged Variable

To calculate the error bars for an averaged variable, on a daily basis or over several weeks, we use the following formula, which applies when the data are independent:

$$\sigma_{\bar{X}} = \frac{1}{\sqrt{N}} \sqrt{\frac{1}{N} \sum_{i=1}^{N} \sigma_i^2} \qquad (B1)$$

with $\sigma_{\bar{X}}$ the standard deviation of the variable $X$, $\sigma_i$ the standard deviation of each element $i$ of $X$ and $N$ the number of elements in $X$.

Figure B1 shows the error bars for the averaged 532 nm CALIOP retrieved AOD. The error bars can be large in the first
few days because we do not take into account orbits with no plume (no retrieval possible, so no associated error) in the error calculation so as not to bias the results, whereas we do take them into account as a zero value in the calculation of the mean in order to make the best possible comparison with the other instruments. Between late January and early March, errors are between 15 and 50%. After, errors increase again as a result of signal attenuation and noise.

*Author contributions.* CD and BL conceived the study. CD, BL and AP developed the code. CD, BL, AP and PS conducted the analyses
and wrote the original draft. AEB and DJZ provided support regarding the usage of the OSIRIS and OMPS-LP USASK data products. AR provided support regarding the usage of the OMPS-LP IUP data product. GT provided support regarding the usage of the OMPS-LP NASA data product. All authors contributed to the discussion and improvement of the initial paper.





*Competing interests.* The authors declare no competing interests.

*Acknowledgements.* This study has been supported by the Agence Nationale de la Recherche under Grant 21-CE01-0007-01 (ANR ASTuS) and the Centre National d'Études Spatiales (CNES) under Grant N° 5100020313 and Grant EXTRA-SAT. OSIRIS operations and data processing are supported by the Swedish National Space Agency and the Canadian Space Agency. The University of Bremen team was funded in part by the European Space Agency (project CREST), by the German Research Foundation (VolImpact, FOR2820), and by the University of Bremen and state of Bremen. The calculations for this study were done at NHR@ZIB and NHR@Göttingen (project hbk00098). Ghassan Taha has been supported by the National Aeronautics and Space Administration, Earth Science Division grant no. 80NSSC23K1037.



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
