# Peer review of "Aerosol Composition and Extinction of the 2022 Hunga Plume Using CALIOP"

_EGUsphere, 2025_

## Author Comment (AC1)

**Reply to referees for the manuscript "Aerosol Composition and Extinction of the 2022 Hunga Plume Using CALIOP"**

We thank the referees for their evaluation of our work and their comments which led us to add several hopefully useful clarifications.

**Modifications:**

- *$Q_s$ was given in $m^2 \cdot mol^{-1}$ in **Section 2.2**, now corrected in $m^2$.*
- *In the caption of **Figure 2**, the DR was defined as the ratio orthogonal/total channel, now corrected as the ratio orthogonal/parallel channel.*
- *We clarified in the **Section 3.1** that the observed change in the lidar ratio is most likely attributable to an evolution of the sulfate aerosol size distribution, and we have explicitly reflected this interpretation in the **Abstract**.*
- ***Figure 5:** addition of **OMPS NASA v2.5**, released publicly in August 2025 (https://doi.org/10.5067/6X1B487WO4W1), after the submission of the initial manuscript. We include this new version in addition to the one already incorporated (**v2.1**), which is still being produced and distributed, so that the article remains fully representative of the datasets currently available, without removing material that is already part of the manuscript.*
- *Addition of a new **appendix** (**B**) with Himawari-8/MSG-1 RGB-Ash maps including CALIOP ground tracks corresponding to panels from Figure 2 following the request of referee #3.*
- *Addition of a new **appendix** (**D**) with some other cases of coincident cases between CALIOP, SAGE III and OMPS-LP, to further support the discussion.*
- *Addition of a new **appendix** (**E**) discussing briefly OMPS-LP NASA filtering, which was revised between versions v2.1 and v2.5.*

**New references:**

- Chen, Z., DeLand, M., and Bhartia, P. K.: A new algorithm for detecting cloud height using OMPS/LP measurements, Atmospheric Measurement Techniques, 9, 1239–1246, https://doi.org/10.5194/amt-9-1239-2016, 2016.
- Kahn, R. A., Limbacher, J. A., Junghenn Noyes, K. T., Flower, V. J. B., Zamora, L. M., and McKee, K. F.: Evolving Particles in the 2022 Hunga Tonga—Hunga Ha'apai Volcano Eruption Plume, Journal of Geophysical Research: Atmospheres, 129, e2023JD039 963, https://doi.org/10.1029/2023JD039963, 2024.
- Taha, G., Loughman, R., Zhu, T., and DeLand, M.: Suomi-NPP OMPS Limb Profiler L2 Aerosol Daily Product, Collection 2.5, Version 2.5, Tech. rep., NASA Goddard Earth Sciences Data and Information Services Center (GES DISC), NASA GSFC Code 610.2, https://omps.gesdisc.eosdis.nasa.gov/data/SNPP_OMPS_Level2/OMPS_NP

P_LP_L2_AER_DAILY.2.5/doc/README.OMPS_NPP_LP_L2_AER_DAILY_v
2.5.pdf, 2025.

**Referee comments:**

Referees' comments are in black and our replies in blue.

Referee #1 :

1. L 67. "the most significant orbit of the day". What is the sensitivity if you take into account e.g. two orbits and not the "most significant"?

[Figure]

*In this new figure, some averaged cases are highlighted in red, corresponding to the two orbits exhibiting the highest mean SR values. On 27 January, the plume is not yet zonally homogenized (and the main part of the plume located in the SAA region at this time), and the values taken from the two orbits differ markedly. As time progresses, the plume becomes increasingly uniform in the zonal direction (on 16 February, the main part of the plume is still located within the SAA region). By 8 March, the plume is fully zonally homogeneous, the AOD and LR values derived*

*from the two orbits are very similar. The error bars are reduced thanks to the use of two orbits instead of a single one.*

    2. L 305-310: The formula calculates the standard error not the standard deviation. So σ(x) should be the standard error.

Corrected : *"standard deviation of the variable X"* → *"standard error of the variable $\bar{X}$"*

Referee #2 :

∅

Referee #3 :

    1. Figure 2 presents 6 CALIOP subplots showing the plume structure with individual clouds (C1 and C2) clearly identified. This figure is very helpful and well described in the main text. However, as lidar provides only a cross-sectional profile of the plume, it is difficult for readers to visualize which part of the overall plume is being sampled. If feasible, I recommend adding an additional panel for each subplot that shows the corresponding CALIOP ground track overlaid on imagery from another instrument (e.g., geostationary satellites or IASI). This would give readers a better sense of the horizontal context and improve interpretability.

Added in appendix (**B**) with Himawari-8/MSG-1 RGB-Ash maps and with references in the main text (not enough space to include new plots in the same figure).

    2. Section 3.2 discusses the evolution of AOD for individual plume features (e.g., C1 and C2). It should be clarified explicitly at the beginning of this section that the reported AOD values refer to plume-layer AOD rather than the total atmospheric column AOD. Since "AOD" typically denotes column-integrated optical depth, it would be helpful to define this clearly here and wherever the concept of plume-layer AOD is used to avoid ambiguity.

Clarification added at the end of section 2.1 "CALIOP" : *"Note that throughout the manuscript, references to CALIOP-derived AOD correspond to the plume-layer AOD, i.e., the aerosol optical depth integrated only over the vertical extent of the detected plume. This differs from the usual column-integrated AOD, but this distinction has little impact on the results because the background extinction outside the plume in the stratosphere is very small compared with the plume extinction."*

3. In Figure 1c, the horizontal section at 17.5 km suggests that the signal between approximately 22°S and 17°S latitude may be fully attenuated. Could the authors comment on how this attenuation might affect the extinction and plume-layer AOD retrievals for the plume near 30 km altitude?

*In Figure 1c, the horizontal cross section at 17.5 km indeed shows ASR values that may suggest locally strong signal attenuation. However, these values — including the two minimum ASR values measured at 14.9°S and 16.7°S — remain strictly positive. When averaging is performed over several degrees of latitude (here 14–18°S) and over several kilometres in altitude, these local minima are smoothed out, and the resulting signal appears less attenuated than what a noisy single-altitude horizontal slice might suggest (see Figure 1f, j). The AOD retrieved for this case (the maximum AOD determined in this study) is consistent with the results shown in Figure 3 of the paper, as well as with AERONET AOD measurements over Australia reported in Boichu et al. (2023).*